# Drug-Coated Balloon versus Plain Balloon Angioplasty in the Treatment of Infrainguinal Vein Bypass Stenosis: A Systematic Review and Meta-Analysis

**DOI:** 10.3390/jcm12010087

**Published:** 2022-12-22

**Authors:** Toshihiko Isaji, Yutaka Hosoi, Kota Kogure, Yohei Ichikawa, Keisuke Fujimaki, Toru Ikezoe, Masao Nunokawa, Hiroshi Kubota

**Affiliations:** Department of Cardiocascular surgery, Kyorin University, 6-20-2, Shinkawa, Mitaka 181-8611, Tokyo, Japan

**Keywords:** drug-coated balloon, infrainguinal artery bypass, bypass stenosis

## Abstract

The optimal endovascular therapy for vein graft stenosis (VGS) following infrainguinal arterial bypass is yet to be established. Drug-coated balloons (DCB) have rapidly improved the inferior patency outcomes of angioplasty using a conventional plain balloon (PB). This study compares the efficacy of DCBs and PBs for the treatment of infrainguinal VGS. This systematic review and meta-analysis was performed according to the PRISMA statement. Multiple electronic searches were conducted in consultation with a health science librarian in September 2022. Studies describing the comparative outcomes of angioplasty using DCBs and PBs in the treatment of infrainguinal VGS were eligible. Datasets from one randomized controlled trial and two cohort studies with a total of 179 patients were identified. The results indicated no significant difference in target lesion revascularization between DCBs and PBs (OR, 0.64; 95% CI, 0.32–1.28; *p* = 0.21), with no significant heterogeneity between studies. Additionally, differences in primary patency, assisted primary patency, secondary patency, and graft occlusion were not significant. Subgroup analysis showed similar effects for different DCB devices. In conclusion, DCBs showed no significant benefit in the treatment of VGS compared to PBs. Given the small population size of this meta-analysis, future trials with a larger population are desired.

## 1. Introduction

Infrainguinal arterial bypass (IAB) using an autologous vein graft is a well-established surgical technique that remains the gold standard for femoropopliteal arterial occlusive disease, particularly for patients with chronic limb-threatening ischemia [1]. In an era of endovascular therapy (EVT), IAB for long arterial occlusion extending to the crural arteries [2], advanced limb threat, or highly complex anatomy resistant to EVT [3] are the best revascularization methods.

Autologous IAB, typically using the great saphenous vein, showed higher longevity (60–80% of five-year patency rates [2,4,5,6]) than IAB with prosthetic grafts. Graft surveillance after autologous IAB occasionally reveals vein graft stenosis (VGS) mostly within two years after the bypass, potentially causing graft occlusion. Regular surveillance and intervention for VGS are essential to further improve long-term graft patency [7,8]. Therapeutic interventions for VGS are classified as surgical and endovascular treatments. Although the efficacy of the former is well established [9,10], the latter is increasingly reported and can be performed less invasively and repeatedly [11]. Angioplasty using balloons is more beneficial than stenting in the absence of a permanent metallic scaffold [12].

Drug-coated balloons (DCB), mostly paclitaxel-coated balloons, are increasingly used in endovascular therapies, improving the inferior patency outcomes of angioplasty with conventional plain balloons (PB). Neointimal hyperplasia (NIH), caused by vascular smooth muscle cell (VSMC) proliferation, is the main etiology of vein graft failure in the intermediate period [13]. Several studies indicate that paclitaxel, a highly lipophilic antineoplastic drug, inhibits the proliferation of VSMC after administration to cell culture and swine tissues for a few minutes [14]. DCBs are valuable in many vascular diseases, including coronary artery disease [15], arteriovenous fistula (AVF) as blood access [16,17], and superficial femoral artery disease [18]. However, the superiority of DCB in the treatment of VGS after IAB remains unclear. The objective of this systematic review and meta-analysis is to assess the efficacy of DCBs compared with PBs in the treatment of infrainguinal vein bypass stenosis.

## 2. Materials and Methods

This systematic review and meta-analysis was performed according to the Preferred Reporting Items for Systematic Reviews and Meta-Analyses (PRISMA) statement. The registration number in PROSPERO is CRD42022352017.

### 2.1. Search Strategy

A comprehensive search of MEDLINE and the Cochrane Library in consultation with a health science librarian (Y.S.) was conducted electronically from 16–26 September 2022. The complete details of the search strategy are provided in the Appendix A. A combination of search words, including bypass graft, drug-coated balloon, drug-eluting balloon, paclitaxel-coated balloon, peripheral arterial disease, femoropopliteal artery disease, and infrainguinal artery disease, were used to probe for eligible studies.

### 2.2. Study Eligibility and Risk of Bias Assessment

Two independent authors (T.I. (Toshihiko Isaji) and Y.H.) reviewed the search results and selected studies. No publication dates or language restrictions were included. Randomized controlled trials (RCTs) and observational studies describing the comparative outcomes of percutaneous transluminal angioplasty (PTA) using drug-coated balloons and plain balloons in the treatment of infrainguinal VGS were considered for inclusion. Studies that focused on balloon angioplasty for the treatment of native arteries were excluded. Case reports were excluded from the study. Outcomes of target lesion revascularization (TLR) and graft patency during routine follow-up after PTA for vein graft stenosis had to be reported. Full-text articles were reviewed for eligibility (T.I. (Toshihiko Isaji) and Y.H.). The Risk of Bias Assessment Tool for Nonrandomized Studies (RoBANS) was used to assess the risk of bias [19]. Two authors (T.I. (Toshihiko Isaji) and Y.H.) independently assessed the RoB.

### 2.3. Data Extraction and Synthesis

Original data were extracted according to the PRISMA checklist. Two independent authors (T.I. (Toshihiko Isaji) and Y.H.) reviewed the eligible data for analysis. Patient demographics, operative characteristics, TLR, primary patency (PP), assisted primary patency (APP), secondary patency (SP), and graft occlusion were obtained from each study. For the systematic review, outcomes were extracted to estimate the odds ratios (OR). If the study reported outcomes using only the Kaplan–Meier curve, the numerical results at certain time points were considered as previously reported [20].

### 2.4. Study Outcomes

The outcomes were assessed in accordance with the reporting standards of the Society for Vascular Surgery [21]. To assess the validity of drug-coated balloons, TLR and graft occlusion in each study were incorporated. Additionally, PP, APP, and SP at one year were assessed. TLR was defined as any revascularization of the same stenotic lesion. PP was expressed as uninterrupted patency after revascularization without additional procedures. APP was defined as the patency achieved with any additional intervention to prevent bypass occlusion. SP was described as the patency achieved with any additional intervention to restore patency after occlusion. A subgroup analysis was performed to assess the outcome differences between the DCB devices.

### 2.5. Statistical Analysis

RevMan version 5.4.1 (Nordic Cochrane Centre, the Cochrane Collaboration, 2020, Copenhagen, Denmark) software was used to incorporate the pooled rates and ORs from each study with 95% CI using the random effects model. Any results with an *I*^2^ > 50% or the *p* value for heterogeneity < 0.05 were considered to represent significant heterogeneity between the studies. A random-effects model, which accounts for the variance between studies, was used to analyze the pooled data. Publication bias was assessed using funnel plot asymmetry. A leave-one-out sensitivity analysis was conducted with the exclusion of one study to confirm that our findings were not driven by any single study.

## 3. Results

### 3.1. Search Results

We initially screened 149 articles (131 from MEDLINE (Washington, DC, USA.) and 18 from the Cochrane Library (London, United Kingdom)). After excluding five duplicate records and 134 records for other reasons, 107 records of coronary artery disease, 10 records of other endovascular therapies, 2 records comparing bypass surgery and angioplasty for VGS, 11 literature reviews, and 4 case reports, 10 articles were screened for their abstracts. Three irrelevant articles and one article that assessed identical data as other papers were excluded. Six articles were screened for full text. We identified three titles, including one RCT and two cohort studies, with a total of 179 patients meeting the inclusion criteria [22,23,24], after excluding the following three papers. The first study compared the outcomes of angioplasty for both failing peripheral arterial vein grafts and synthetic bypass grafts; the data of the vein graft group could not be extracted [25]. The second study was a single-arm study of drug-coated balloon treatment for infrainguinal bypass graft stenosis, including both vein grafts and prosthetic grafts [26]. The third study showed severe bias in the selection of participants. Each patient belonged to both the drug-coated balloon group and the plain balloon group simultaneously, as only restenosis or occlusion in vein grafts that had been previously treated with plain balloons was indicated for drug-coated balloon treatment [27]. The PRISMA flow diagram for systematic reviews is depicted in Figure 1.

### 3.2. Study Characteristics

Of the 179 patients, 92 (51%) were treated with DCBs, and 87 (49%) were treated with PBs. The baseline characteristics of the three studies are shown in Table 1. The treatment characteristics of angioplasty for vein graft stenosis are presented in Table 2. The follow-up methods and indications for the intervention are depicted in Table 3. In all studies, duplex ultrasound was performed when symptomatic or regularly after arterial bypass, and angioplasty was considered for graft stenosis that was defined by a peak systolic velocity or peak systolic velocity ratio on duplex ultrasound. All DCBs used in the studies were coated with paclitaxel at a dose of 3.0 µg/mm^2^ [21] or 3.5 µg/mm^2^ [22,24] on the balloon surface. The balloon diameters of the DCBs and PBs were similar. The treatment location was described in two studies [22,23]. As shown in Figure 2, the RoBANS assessment demonstrated a high risk of bias regarding the participant selection (33%) and confounding variables (33%).

### 3.3. Outcomes

Two studies reported TLR at one year. TLR during the whole study period was reported in one study, of which the mean follow-up was 2.5 years [22]. The difference in TLR between DCBs and PBs was not significant (OR, 0.64; 95% CI, 0.32–1.28; *p* = 0.21) (Figure 3A), with no significant heterogeneity between studies (*I*^2^ = 0%; *p* = 0.94). Two studies reported the outcomes of PP at one year. The difference in PP between DCBs and PBs was not significant (OR, 1.26; 95% CI, 0.54–2.94; *p* = 0.59; *I*^2^ = 0%) (Figure 3B), and there was no significant heterogeneity between studies (*I*^2^ = 0%; *p* = 0.42). Three studies reported the outcomes of APP at one year. The difference in APP between DCBs and PBs was not significant (OR, 1.55; 95% CI, 0.55–4.34; *p* = 0.40) (Figure 3C), without significant heterogeneity between studies (*I*^2^ = 0%; *p* = 0.74). Three studies reported the outcomes of SP at one year. The difference in SP between DCBs and PBs was not significant (OR, 1.45; 95% CI, 0.44–4.79; *p* = 0.54) (Figure 3D), with no significant heterogeneity between studies (*I*^2^ = 0%; *p* = 0.46). Three studies reported the outcomes of graft occlusion. The difference in graft occlusion between DCBs and PBs was not significant (OR, 0.65; 95% CI, 0.25–1.68; *p* = 0.38) (Figure 3E), and there was no significant heterogeneity between studies (*I*^2^ = 0%; *p* = 0.56). Subgroup analysis classifying the studies according to the DCB devices showed similar effects of DCB in studies using InPact and Passeo-19 Lux at one-year assisted primary patency. Funnel plots revealed no evidence of publication bias (Figure 4).

## 4. Discussion

The results of the systematic review indicate that DCBs are not significantly superior to PBs in reducing the occurrence of TLR in the treatment of infrainguinal VGS, and no significant differences were observed in the graft patency of PP, APP, and SP at one year and graft occlusion. Additionally, the one-year APP was not significantly different between the two DCB devices, and no significant heterogeneity was observed in any of these outcomes.

No meta-analysis or multicenter trial comparing the outcomes of DCB and PB in the endovascular treatment of infrainguinal vein bypass stenosis has been reported previously as far as we know. Two cohort studies and one randomized controlled study were included in this review. In the RoBANS assessment, one study was considered to have a high risk of participant selection bias. In this study, consecutive patients with VGS and treated with DCBs were prospectively enrolled after the treatment regimen of VGS had changed from PBs. Data from patients treated with PBs before the treatment regimen change were retrospectively collected [23]. The other study presented a high risk of bias in confounding variables because significantly more patients presented Rutherford category 2 ischemia in the PB group than in the DCB group [22].

In contrast to this meta-analysis, previously reported studies concerning the EVT of other vascular diseases showed the benefit of DCBs in preventing the restenosis of target lesions compared to PBs [15,16,17,18]. Among them, AVF has similar aspects to vein grafts because both require venous adaptation to the arterial environment to achieve a long-term duration. Venous adaptation integrates outward remodeling and wall thickening. Vein grafts adapt mainly via increased wall thickening with less outward dilation, whereas AVF adapts mainly through outward dilation and less intimal thickening [28]. Venous wall thickening leading to stenosis in both AVF and vein grafts is mainly caused by NIH in the venous wall, which typically occurs within the first year, which was consistent with the observational periods of this meta-analysis. DCB inhibits the proliferation of VSMC after intimal injury by delivering an antiproliferative drug with an excipient substance that promotes drug transfer to the vessel wall [29]. However, progressive NIH in vein graft stenosis may reduce the effect of paclitaxel on DCB, which inhibits VSMC proliferation.

In addition, the mechanisms of vein graft failure appear more complex than those of VSMC proliferation [30,31]. PREVENT trials have been performed to overcome vein graft failure by regulating VSMC proliferation. E2F decoy, which blocks the activation of genes mediating cell cycle progression and intimal hyperplasia, successfully inhibited VSMC proliferation in a rat carotid injury model [32]. However, the PREVENT-III study showed no predominance of the patient group treated with the E2F decoy in time to graft reintervention or major amputation, suggesting that strategies inhibiting VSMC growth do not prevent vein graft stenosis. Furthermore, technical errors in anastomoses, graft routes, or intimal injuries during graft harvesting can cause VGS [33].

Drug-eluting stents (DES), including paclitaxel-eluting stents, are reportedly associated with a small increase in stent thrombosis after percutaneous coronary interventions compared with bare-metal stents [34]. Double antiplatelet therapy (DAPT) after DES placement or DCB angioplasty has been widely accepted. The studies included in this meta-analysis did not strictly follow the DAPT regimen; instead, they adopted different regimens of postprocedural antithrombotic medication. This may have prevented DCB angioplasty from achieving superior results in terms of inhibiting restenosis and repeat angioplasty.

Hence, we need to further explore the treatment options to improve the outcomes of VGS. DES therapy for the treatment of infrainguinal VGS has better primary patency than cutting balloon and plain balloon angioplasty [35]. However, external forces on stents due to the anatomic condition of vein grafts, such as subcutaneous tunneled routes and crossing of joint space, may cause stent fracture. In contrast, the advantage of DCB therapy is that it does not require a permanent metallic implant, thus preserving the subsequent treatment options. As such, VGS lesions suitable for DES therapy may be limited. Further studies comparing DESs and DCBs in the treatment of VGS are required to assess the benefit of non-metallic implants. In addition, circulating biomarkers as predictors of postinterventional restenosis for VGS could help optimize the use of DCBs. For instance, proinflammatory markers and high plasma levels of micro-RNA have been reported to potentially influence the outcomes after peripheral stenting [36].

This meta-analysis had several limitations. First, except for one RCT, two of the three included studies were retrospective cohort studies, leading to a low level of evidence. Second, each study was underpowered by sample size. Therefore, a lack of statistically significant difference does not mean no difference between the two groups. As EVT therapy is first recommended for most lesions of infrainguinal artery diseases, it is difficult to increase the entry of VGS cases after IAB in additional studies. Third, the treatment methods of VGS after IAB intrinsically include complexity and heterogeneity between studies; especially with regard to IAB procedures such as anastomosis fashion and grafting route as well as EVT procedures such as the predilatation method. As mentioned above, the included studies adopted different regimens of post-procedural antithrombotic medication.

## 5. Conclusions

Comparing DCB with PB angioplasty in the treatment of VGS, a meta-analysis showed no significant difference in outcomes with regard to TLR and graft patency at one year. No significant heterogeneity between the studies was observed. Given the small population of this meta-analysis and the ever-evolving nature of endovascular techniques and devices, future trials with a large population from multiple institutions are required to change or reinforce the results.

## Figures and Tables

**Figure 1 jcm-12-00087-f001:**
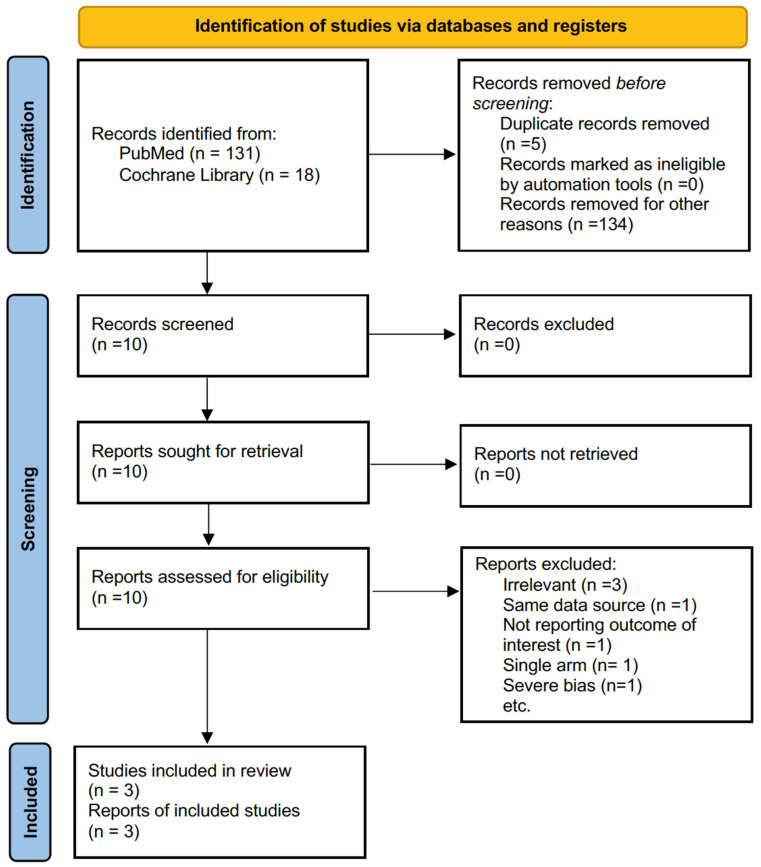
Preferred reporting items for systematic reviews and meta-analyses diagram for article selection.

**Figure 2 jcm-12-00087-f002:**
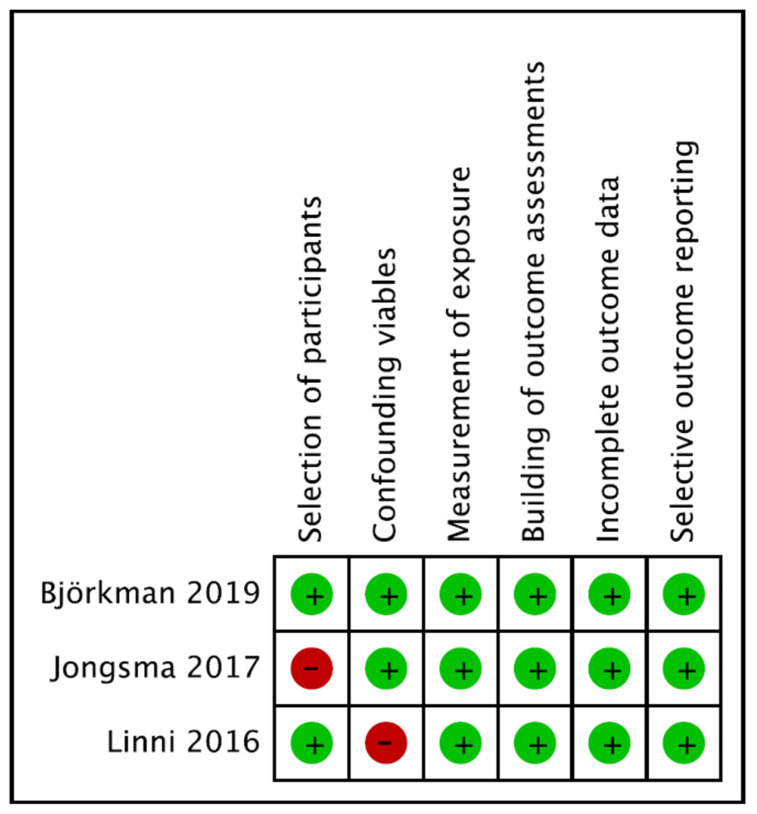
Risk of Bias Assessment Tool for Nonrandomized Studies (RoBANS) [22,23,24]. + in green circle meants “High”. − in red circle means “Low”.

**Figure 3 jcm-12-00087-f003:**
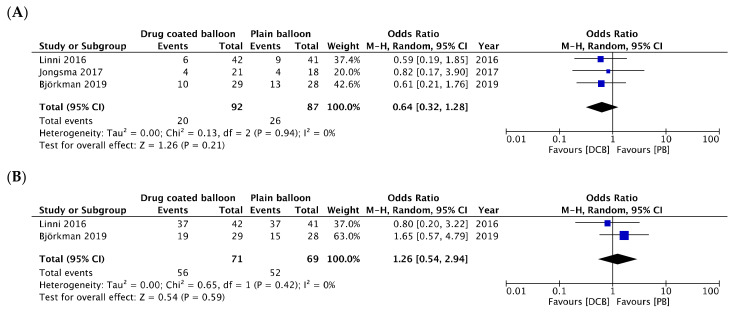
Forest plots of the meta-analysis results of drug-coated balloons (DCB) vs. plain balloons (PB). Angioplasty in studies. (**A**). Target lesion vascularization. (**B**). One-year primary patency. (**C**). One-year assisted primary patency. (**D**). One-year secondary patency. (**E**). Graft occlusion. (**F**,**G**). Subgroup analysis of one-year assisted primary patency according to DCB device. (**F**). In.Pact. (**G**). Passeo-18 Lux. M-H, Mantel-Haenszel; CI, confidence interval; I2, study heterogeneity; Z, *Z*-score test statistic [22,23,24].

**Figure 4 jcm-12-00087-f004:**
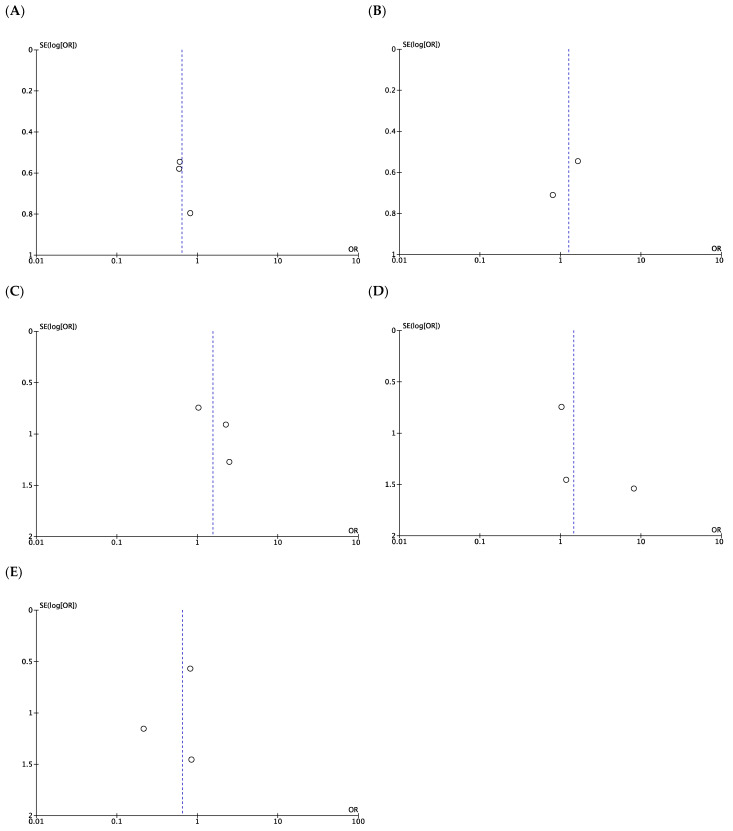
Funnel plots of the meta-analysis results of drug-coated balloons (DCB) vs. plain balloons (PB) angioplasty in studies. (**A**). Target lesion vascularization. (**B**). One-year primary patency. (**C**). One-year assisted primary patency. (**D**). One-year secondary patency. (**E**). Graft occlusion.

**Table 1 jcm-12-00087-t001:** Baseline characteristics.

First Author, Year		Linni, 2016 [22]	Jongsma, 2017 [23]	Björkman, 2019 [24]
Country		Austria	Netherlands	Finland
Study design		CT	CT	RCT
Patients (No.) (total)		42 vs. 41 (83)	21 vs. 18 (39)	29 vs. 28 (57)
Mean age (years)		70 vs. 71	70 vs. 71	70 vs. 72
Male (%)		21 vs. 39	62 vs. 39	61 vs. 52
DM (%)		40 vs. 46	29 vs. 28	38 vs 61 *
Smokers (%)		36 vs. 34	81 vs. 78	59 vs. 50
HTN (%)		95 vs. 95	57 vs. 83	79 vs. 71
DL (%)		55 vs. 63	52 vs. 78	90 vs. 96
CKD (%)		NA	10 vs. 0	21 vs. 14
CAD (%)		55 vs. 56	48 vs. 39	48 vs. 39
CLTI before PTA (%)		26 vs. 42	10 vs. 17	55 vs. 36
pre ABI		0.57 vs. 0.51	NA	0.6 vs. 0.74
Bypass distal anastomosis (%)	Above knee popliteal	33 vs. 37	76 vs. 89	14 vs. 14
	Below knee popliteal	67 vs.63		38 vs. 46
	crural		24 vs. 11	48 vs. 39
Bypass with spliced vein graft (%)		NA	14 vs. 11	31 vs. 32

CT, cohort study; RCT, randomized control study; DCB, drug-coated balloon; PB, plain balloon; DM, diabetes mellitus; HTN, hypertension; DL, dyslipidemia; CKD, chronic kidney disease; CAD, coronary artery disease; CLTI, chronic limb-threatening ischemia; PTA, percutaneous transluminal angioplasty; ABI, ankle-brachial pressure index; NA, not available. Each column shows the comparison data of the DCB and PB. * Significant difference (*p* = 0.012).

**Table 2 jcm-12-00087-t002:** Treatment characteristics.

First Author, Year	Linni, 2016 [22]	Jongsma, 2017 [23]	Björkman, 2019 [24]
DCB balloon	In.Pact	Passeo-18 Lux	In.Pact
Balloon diameter (DCB vs. PB; mm)	NA	3.8 vs. 3.7	4.9 vs. 5.3

DCB, drug-coated balloon; PB, plain balloon; NA, not available.

**Table 3 jcm-12-00087-t003:** Follow-up methods and indications for intervention.

First Author, Year	Linni, 2016 [22]	Jongsma, 2017 [23]	Björkman, 2019 [24]
Indication for PTA	>70% bypass stenosis verified as a PSV < 45 cm/s or 300 cm/s or PSVR > 4 in DUS imaging.	>70% bypass stenosis verified as a PSV > 300 cm/s or PSVR > 3.0 in DUS imaging.	PSVR > 2.5.
Follow-up protocol	DUS was performed in case of recurrent symptoms or decrease of ABI > 0.15 at 1, 3, 6, and 12 months after PTA and at 12-month intervals thereafter. Reintervention was performed for restenosis of >70%.	DUS was performed at 6 and 12 months after PTA. Reintervention was performed for restenosis of >70%.	DUS and clinical evaluation (symptoms, ABI) were performed at 1, 6, and 12 months after PTA. Reintervention was performed for PSVR >2.5
Antithrombotic medication (AM)	No specific regimen. Aspirin 100 mg for patients taking no AM at admission.	Anticoagulants or aspirin for at least 2 years after bypass surgery.	aspirin 100 mg + clopidogrel 75 mg or warfarin + aspirin 50 mg for three months for all patients

PTA, percutaneous transluminal angioplasty; DUS, Doppler ultrasonography; PSV, peak systolic velocity; PSVR, peak systolic velocity ratio.

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
