# Peer review of "Drug-Coated Balloon versus Plain Balloon Angioplasty in the Treatment of Infrainguinal Vein Bypass Stenosis: A Systematic Review and Meta-Analysis"

_jcm, 2022, doi:10.3390/jcm12010087_

Round 1
Reviewer 1 Report
Dear Authors,
I am pleased to review this manuscript, which focuses on the role of DCBs in the maintenance of vein bypass graft. My opinion is that it is an interesting and original topic that needs more discussion and analysis. A good review of the literature was performed, but a little improvement could be made by adding newest references. Some points are missing in the present version of the manuscript that could greatly improve its structure and usefulness.
First of all, four figures are mentioned in the manuscript, but no figures are present in the downloadable manuscript.
The introduction is well structured. In the era of endovascular therapy, bypass surgery is re-establishing the first choice for the treatment of critical limb ischemia. I believe the authors should emphasize this validation. Citing Enlg J Med 2022. Nov 7. doi:10.1056/NEJMoa2207899 might help the discussion.
One of the substantial limitation of the present study is the small number of selected studies.
Line 118-119: Why did the authors exclude
-
San Norberto E.M.; Taylor J.H.; Carrera S.; Vaquero C. Percutaneous Transluminal Angioplasty With Drug-Eluting Balloons for 325 Salvage of Infrainguinal Bypass Grafts.J. Endovasc. Ther. 2014, 21, 12–21; doi: 10.1583/13-4473R.1.
from the analysis? Couldn't the results provided by this single-arm study be added to the other analyzed data?
Discussion is well conducted. However, cite recent studies [i.e.: Cardiovasc Intervent Radiol 44, 207–217 (2021); Diagnostics (Basel). 2022 Sep 12;12(9):2207. doi: 10.3390/diagnostics12092207; Ann Vasc Surg. 2022 Jul 8:S0890-5096(22)00325-9. doi: 10.1016/j.avsg.2022.06.006]. that validate the use of the DCB's and the drug-coated technology in general, in the treatment of PAD and in particular of CLI, can help make the discussion more interesting, increasing the appeal of the entire manuscript.
Author Response
First of all, four figures are mentioned in the manuscript, but no figures are present in the downloadable manuscript.
Response: I believe you would find the four figures in the middle of this manuscript.
The introduction is well structured. In the era of endovascular therapy, bypass surgery is re-establishing the first choice for the treatment of critical limb ischemia. I believe the authors should emphasize this validation. Citing Enlg J Med 2022. Nov 7. doi:10.1056/NEJMoa2207899 might help the discussion.
Response: Thank you for your recommendation. We emphasized the validation of bypass surgery for the treatment of CLTI by citing this paper, Enlg J Med 2022. Nov 7. doi:10.1056/NEJMoa2207899.
One of the substantial limitation of the present study is the small number of selected studies.
Line 118-119: Why did the authors exclude
- San Norberto E.M.; Taylor J.H.; Carrera S.; Vaquero C. Percutaneous Transluminal Angioplasty With Drug-Eluting Balloons for 325 Salvage of Infrainguinal Bypass Grafts.J. Endovasc. Ther. 2014, 21, 12–21; doi: 10.1583/13-4473R.1.
from the analysis? Couldn't the results provided by this single-arm study be added to the other analyzed data?
Response: I totally agree with your opinion. The small number of studies is the substantial limitation of this systematic review. We excluded the study “Percutaneous Transluminal Angioplasty With Drug-Eluting Balloons for 325 Salvage of Infrainguinal Bypass Grafts”, not only because this is a single-arm study, but because this study includes both failing peripheral arterial vein grafts and synthetic bypass grafts and the data of the vein graft group could not be extracted. We addressed this point in this manuscript.
Discussion is well conducted. However, cite recent studies [i.e.: Cardiovasc Intervent Radiol 44, 207–217 (2021); Diagnostics (Basel). 2022 Sep 12;12(9):2207. doi: 10.3390/diagnostics12092207; Ann Vasc Surg. 2022 Jul 8:S0890-5096(22)00325-9. doi: 10.1016/j.avsg.2022.06.006]. that validate the use of the DCB's and the drug-coated technology in general, in the treatment of PAD and in particular of CLI, can help make the discussion more interesting, increasing the appeal of the entire manuscript.
Response: Thank you for your recommendation. We cited a novel study validating the optimal use of DCB in line 253-256.
Reviewer 2 Report
The management of restenosis after infrainguinal vein bypass is interesting.
Also, the choice of the preferred treatment for vein graft restenosis (surgical vs endovascular: DCB or PB) is still debated.
The review is interesting and well conducted, however, there are same criticisms
Specific consideration:
- Lines 39-40, I suggest adding proper reference.
- Line 111, I suggest specifying, if possible, the main reasons for the exclusion of 134 studies “for other reasons”
- Line 118: could the authors explain the decision to exclude a single-arm study of drug-coated balloon treatment for infrainguinal bypass?ref [24].
Author Response
Lines 39-40, I suggest adding proper reference.
Response: We cited a paper addressing this theory.
Line 111, I suggest specifying, if possible, the main reasons for the exclusion of 134 studies “for other reasons”
Response: We addressed the main reasons for the exclusion of 134 studies “for other reasons” in line 112- 114.
Line 118: could the authors explain the decision to exclude a single-arm study of drug-coated balloon treatment for infrainguinal bypass?ref [24].
Response: We excluded the study “Percutaneous Transluminal Angioplasty With Drug-Eluting Balloons for 325 Salvage of Infrainguinal Bypass Grafts”, not only because this is a single-arm study, but because this study includes both failing peripheral arterial vein grafts and synthetic bypass grafts and the data of the vein graft group could not be extracted. We addressed this point in this manuscript.
Reviewer 3 Report
Line 41: Instaed of 'are rapidly used ...: 'are increasingly used' ...
General: As the authors mention the analysis suffers from the unavoidably small number of patients and trials. The authors should not suggest that 'lack of statistival significance' means no difference. It may be simply due to the small numbers.
Considering the complexity of patients, surgical and endovascular methods the meta-analysis is underpowered to decide the question if DCB improve the outcome of treatment over plain angioplasty.
Author Response
Line 41: Instaed of 'are rapidly used ...: 'are increasingly used' ...
Response: We revised it as you recommended.
General: As the authors mention the analysis suffers from the unavoidably small number of patients and trials. The authors should not suggest that 'lack of statistival significance' means no difference. It may be simply due to the small numbers.
Response: We totally agree with your opinion, and mentioned the limitation of this point in line 254-257.
Considering the complexity of patients, surgical and endovascular methods the meta-analysis is underpowered to decide the question if DCB improve the outcome of treatment over plain angioplasty.
Response: Revised accordingly in line 263.